# Biological Roles and Pathogenic Mechanisms of LncRNA MIR4435-2HG in Cancer: A Comprehensive Review

Zhou Chen [1,2], Defeng Guan [1,2], Qiangping Zhu [1], Zhengfeng Wang [1,2], Fangfang Han [1,2] and Wence Zhou [1,3,*]

1   The First Clinical Medical College, Lanzhou University, Lanzhou 730000, China
2   The First Hospital of Lanzhou University, Lanzhou 730000, China
3   The Department of General Surgery, Lanzhou University Second Hospital, Lanzhou 730000, China
*   Correspondence: zhouwc129@163.com or chenzh20@lzu.edu.cn

**Abstract:** The long non-coding RNA MIR4435-2HG has been confirmed to play a crucial regulatory role in various types of tumors. As a novel type of non-coding RNA, MIR4435-2HG plays a key role in regulating the expression of tumor-related genes, interfering with cellular signaling pathways, and affecting tumor immune evasion. Its unique structure allows it to regulate the expression of various tumor-related genes through different pathways, participating in the regulation of tumor signaling pathways, such as regulating the expression of oncogenes and tumor suppressor genes, influencing the biological behaviors of proliferation, metastasis, and apoptosis in tumors. Numerous studies have found a high expression of MIR4435-2HG in various tumor tissues, closely related to the clinical pathological characteristics of tumors, such as staging, lymph node metastasis and prognosis. Some studies have discovered that MIR4435-2HG can regulate the sensitivity of tumor cells to chemotherapy drugs, affecting tumor cell drug resistance. This provides new insights into overcoming tumor drug resistance by regulating MIR4435-2HG. Therefore, studying its molecular mechanisms, expression regulation, and its relationship with the clinical features of tumors is of great significance for revealing the mechanisms of tumor occurrence and developing new therapeutic targets.

**Keywords:** long non-coding RNA; MIR4435-2HG; cancer; tumor microenvironment

## 1. Introduction

Cancer poses a significant health obstacle for the contemporary world, greatly affecting the well-being of the general population. Cancer rates and fatalities are steadily on the rise, making it a leading contributor to global mortality and greatly impacting public health systems and socio-economic factors [1]. The incidence of cancer is showing a continuous upward trend. There are various reasons for the high occurrence of cancer, such as a growing population of older individuals, unhealthy choices in lifestyle, pollution in the environment, inadequate dietary practices, and the presence of chronic illnesses. Cancer occurrence varies considerably across diverse regions and populations, yet in general, the prevalence of cancer is progressively escalating with each passing year [2]. Additionally, the fatality rate attributed to cancer is also experiencing an upward trend. With the rise in cancer cases, there is also a corresponding increase in the number of fatalities caused by this disease. The high mortality rate from cancer imposes a heavy burden on individuals, families, and society. Especially for tumors with a considerable level of seriousness and restricted therapeutic alternatives, the outlook for patients is frequently discouraging [3]. Cancer poses substantial obstacles to the well-being of the population. Patients with cancer necessitate extended, intricate, and costly therapies, imposing a significant burden on healthcare resources. Furthermore, cancer has a profound effect on the well-being of individuals, both physically and mentally, and also affects their familial and social connections.

Long non-coding RNAs (lncRNAs) are RNA molecules that are longer than 200 nucleotides. Unlike messenger RNAs (mRNAs), they do not encode proteins. These

molecules have important functions in gene regulation, impacting gene expression through different mechanisms such as chromatin remodeling, post-transcriptional regulation, and transcription factor activity [4–6]. MIR4435-2HG, also known as LINC00978, AK001796, MORRBID, and lncRNA-AWPPH, is a lncRNA situated on chromosome region 2q13, comprising 10 exons [7,8]. Research suggests that its increased expression in different types of cancers is associated with unfavorable prognosis [9–13]. The underlying mechanism involves the interaction of MIR4435-2HG with microRNAs (miRNAs) and proteins, thereby influencing cellular functions. For instance, MIR4435-2HG has the ability to stimulate the growth of liver cancer cells by upregulating miRNA-487a [14]. Furthermore, it affects the advancement of bladder cancer by regulating cell cycle factors and the signaling of mammalian target of rapamycin (mTOR) [15]. MIR4435-2HG is also involved in glycolysis, epithelial-mesenchymal transition (EMT), and the infiltration of immune cells in cancer [16]. Additionally, MIR4435-2HG has the potential to enhance the polarization of M1 to M2 macrophages, increase macrophage migration, elevate the expression of programmed cell death protein 1 (PD-1), programmed death-ligand 1 (PD-L1), and cytotoxic T-lymphocyte-associated antigen-4 (CTLA4), suppress the immune response of CD8$^+$ T cells, and stimulate the growth and migration of breast cancer cells [17]. However, certain experimental findings suggest that MIR4435-2HG might function as a tumor suppressor gene. The absence of MIR4435-2HG results in a reduction in neutrophils and an elevation in polymorphonuclear myeloid-derived suppressor cells (PMN-MDSC), which hinders PMN-MDSC's metabolism of fatty acids and exerts immunosuppressive effects [18].

Utilizing the oncogenic potential of MIR4435-2HG in various cancers, we employ short interfering RNA (siRNA) and antisense oligonucleotides (ASO) to facilitate oncogene suppression and reduce the expression of target gene proteins, thereby yielding therapeutic benefits [19,20]. Considering the intricate spatial structures and multifaceted biological functions of lncRNAs in cancer therapy, lncRNAs have arisen as promising drug targets for modulating several small molecules [21]. Nevertheless, the development of small molecule drugs targeting lncRNAs is still in the pre-clinical research phase, and there have been no reported instances of small molecule drugs targeting lncRNAs advancing to clinical trials. Regardless, investigating the correlation between MIR4435-2HG and cancer can assist in gaining a more profound comprehension of disease mechanisms and offering hints for innovative therapeutic methods.

## 2. The Role of MIR4435-2HG in the Advancement and Prediction of Cancer

According to extensive findings, MIR4435-2HG displays an atypical pattern of expression in various cancers (Table 1). Moreover, its abnormal expression is closely associated with promoting the biological characteristics of tumor cells. This lncRNA plays a dominant role in the malignant behaviors of tumor cells, such as proliferation, invasion, migration, and metastasis. Particularly, it may serve as a potential prognostic biomarker for patients, especially in promoting the EMT process of tumor cells [7,22].

**Table 1.** Studies related to MIR4435-2HG in various cancers.

| Cancer Type | MIR4435-2HG Expression | Malignant Behaviors | Prognosis | References |
| --- | --- | --- | --- | --- |
| Gastric cancer | Up-regulated | Enhanced cancer cell proliferation, invasion and migration. | Poor | [7,23–26] |
| Hepatocellular carcinoma | Up-regulated | Enhanced cancer cell proliferation, invasion, migration and EMT. | Poor | [14,27–29] |
| Cholangiocarcinoma | Up-regulated | Promoted cancer cell proliferation and migration. | Poor | [30] |
| Colorectal cancer | Up-regulated | Promoted cancer cell proliferation, invasion and migration. | Poor | [11,31–35] |
| Lung cancer | Up-regulated | Enhanced cancer cell proliferation, invasion, migration and EMT. | - | [8,36] |

Table 1. *Cont.*

| Cancer Type | MIR4435-2HG Expression | Malignant Behaviors | Prognosis | References |
|---|---|---|---|---|
| Breast cancer | Up-regulated | Enhanced cancer cell proliferation, invasion, migration and EMT. | Poor | [9,37–39] |
| Glioma | Up-regulated | Enhanced cancer cell proliferation, invasion, migration and EMT. | - | [40–43] |
| Head and neck squamous cell carcinoma | Up-regulated | Enhanced cancer cell proliferation, invasion, migration and EMT. | Poor | [44–49] |
| Ovarian cancer | Up-regulated | Promoted cancer cell proliferation, invasion and migration. | - | [50,51] |
| Cervical cancer | Up-regulated | Promoted cancer cell proliferation, invasion and migration. | - | [10] |
| Clear cell renal cell carcinoma | Up-regulated | Promoted cancer cell proliferation and migration. | - | [52] |
| Bladder cancer | Up-regulated | Promoted cancer cell proliferation, invasion and migration. | - | [53,54] |
| Prostate cancer | Up-regulated | Promoted cancer cell proliferation, invasion and migration. | Poor | [55,56] |
| Melanoma | Up-regulated | Promoted cancer cell proliferation and migration. | - | [57] |

### 2.1. MIR4435-2HG and Gastric Cancer

Gastric cancer, a malignancy that arises from the tissue of the stomach, is widely recognized as a prevalent form of cancer worldwide. Gastric cancer has a high occurrence rate in certain Asian countries such as China, Japan, and South Korea, making it the fourth most prevalent cancer and the second leading cause of cancer-related deaths worldwide [58,59]. According to data from The Cancer Genome Atlas (TCGA) database, MIR4435-2HG is significantly increased in gastric cancer tissues and is closely linked to the overall survival of patients [60]. Additionally, studies have shown a notable rise in MIR4435-2HG in the tissues, plasma, and cell lines of gastric cancer patients, which is associated with tumor size, lymph node metastasis, and TNM staging. This suggests its potential as a molecular biomarker for diagnosing or predicting the prognosis of gastric cancer patients [7,23,24]. In vitro experiments have demonstrated that suppressing MIR4435-2HG inhibits the growth and invasion of gastric cancer cells, and in vivo studies have confirmed its ability to hinder gastric cancer growth [12]. Moreover, a high expression of MIR4435-2HG is inversely correlated with the survival rate of gastric cancer patients. The viability and migration capability of gastric cancer cells are decreased by inhibiting the function of MIR4435-2HG, indicating the important role of MIR4435-2HG in the development of gastric cancer and its potential value in therapy [25,26].

### 2.2. MIR4435-2HG and Hepatocellular Carcinoma

Globally, liver cancer is the sixth most frequently detected cancer and ranks third in terms of cancer-related fatalities [61]. Among all liver cancers, hepatocellular carcinoma is the most prevalent type, originating from liver cells and accounting for 85% of all primary malignant liver tumors [62]. It is concerning that in 2020, hepatocellular carcinoma resulted in more than 830,180 deaths worldwide, positioning it as the third leading cause of cancer-related fatalities [63]. Studies suggest that MIR4435-2HG exhibits a significant increase in hepatocellular carcinoma tissues compared to adjacent normal tissues and shows a positive correlation with tumor size, metastasis, and unfavorable prognosis, and can be used as a prognostic biomarker [64]. Additionally, studies have found that silencing MIR4435-2HG inhibits the migration ability and EMT process induced by hepatic stellate cells in hepatocellular carcinoma cells [14,27,28]. Transcriptomics analysis revealed a substantial increase in the relative expression of MIR4435-2HG in hepatocellular carcinoma cohorts when compared to non-hepatocellular carcinoma and healthy control cohorts. High levels

of MIR4435-2HG expression are strongly linked to reduced overall survival in patients with hepatocellular carcinoma, indicating its potential as a biomarker for this type of cancer. According to functional enrichment analysis, MIR4435-2HG is mainly linked to the development of cancer and the immune system's response [29]. Additionally, it is suggested that MIR4435-2HG could interact with EZH2, causing an increase in its presence in the promoter areas of p21 and E-cadherin genes. This interaction ultimately results in H27K3 trimethylation. In the meantime, the progression of hepatocellular carcinoma is facilitated by the inhibition of p21 and E-cadherin expression by MIR4435-2HG [65]. Additionally, there could be a correlation between MIR4435-2HG and immune infiltration as well as immune checkpoint blockade therapy in hepatocellular carcinoma [66]. Lastly, MIR4435-2HG might play a role in the pyroptosis process and act as a separate prognostic indicator for patients with hepatocellular carcinoma [67].

### 2.3. MIR4435-2HG and Cholangiocarcinoma

Cholangiocarcinoma, the second most prevalent primary liver malignancy after hepatocellular carcinoma, is a cancerous tumor that originates from the biliary epithelium [68,69]. Bioinformatics analysis has revealed MIR4435-2HG as a noteworthy lncRNA. Research conducted using laboratory and animal models has demonstrated that MIR4435-2HG has the ability to enhance the advancement of cholangiocarcinoma. Hence, MIR4435-2HG has the potential to function as a new predictive indicator and target for the treatment of cholangiocarcinoma [30].

### 2.4. MIR4435-2HG and Colorectal Cancer

Globally, colorectal cancer ranks as the third most prevalent form of cancer. The frequency of its occurrence is continuously increasing with the growing elderly population [70]. The expression levels of MIR4435-2HG in colorectal cancer tissues and the serum of colorectal cancer patients are significantly elevated compared with normal colon tissues and serum. The upregulation of MIR4435-2HG is closely linked to the size of tumors, the spread of cancer cells to lymph nodes, TNM staging, progression-free survival, and overall survival, indicating its potential as a biomarker for diagnosing and predicting colorectal cancer [11,31–33,71]. The expression level of MIR4435-2HG is strongly associated with the ability of colon cancer cells to proliferate, adhere, and invade [34,35]. Inhibiting MIR4435-2HG can suppress the proliferation, invasion, and migration of colorectal cancer cells, as well as inhibit the growth of colorectal cancer and its spread to the liver in laboratory settings [11]. Moreover, MIR4435-2HG may play a role in ferroptosis and the infiltration of immune cells [72]. Additionally, research has shown that MIR4435-2HG enhances the resistance of colorectal cancer cells to cisplatin by activating the NRF2/HO-1 pathway [73]. These findings offer new insights into the understanding of the function of MIR4435-2HG in the development and treatment of colorectal cancer.

### 2.5. MIR4435-2HG and Lung Cancer

Among all types of cancer, lung cancer is not only one of the most prevalent malignancies but also has the highest fatality rate. There are primarily two categories of lung cancer: small cell lung cancer and non-small cell lung cancer, where non-small cell lung cancer makes up approximately 80% to 85% of all lung cancer cases [63]. Numerous studies have indicated a significant expression of MIR4435-2HG in lung cancer tissues, which is strongly linked to histological grade and the spread of cancer cells to lymph nodes. Increased concentrations of MIR4435-2HG facilitate the growth and infiltration of lung carcinoma cells. On the other hand, when the expression of MIR4435-2HG is decreased, it hinders the growth and invasion of lung cancer cells both in laboratory settings and in living organisms. Additionally, it also suppresses EMT process and the characteristics of cancer stem cells in lung cancer cells [36]. Furthermore, the reduction in MIR4435-2HG has been shown to decrease the abilities of non-small cell lung cancer cells to proliferate and migrate [8]. These research findings highlight the significant role that MIR4435-2HG plays in the development

of lung cancer, providing a basis for further investigation into its potential value in the treatment of this disease.

## 2.6. MIR4435-2HG and Breast Cancer

The occurrence of breast cancer is increasing annually, making it the second most common cause of cancer-related deaths among women. Several studies have indicated a strong association between the elevated expression of MIR4435-2HG and reduced disease-free survival in patients with breast cancer [74,75]. Following a multivariate analysis, it was concluded that MIR4435-2HG acts as an autonomous prognostic element for breast cancer and has the potential to be a biomarker for forecasting the prognosis of breast cancer patients [37,38]. Moreover, MIR4435-2HG might operate as an oncogene in breast cancer and serve as a potential biomarker for predicting the prognosis of breast cancer patients. Studies suggest that the expression of MIR4435-2HG is markedly elevated in breast cancer tissues and cell lines, and strongly linked to the vitality, proliferation, migration, invasion, and EMT progression of breast cancer cell lines [39]. Suppression of MIR4435-2HG leads to heightened apoptosis in breast cancer cells, accomplished by augmenting the levels of apoptosis-associated proteins and diminishing the levels of anti-apoptotic proteins. Moreover, the suppression of MIR4435-2HG hampers the growth, movement, and infiltration of cells [9]. These study results emphasize the crucial function of MIR4435-2HG in the progression of breast cancer and establish a foundation for its possible efficacy in breast cancer therapy.

## 2.7. MIR4435-2HG and Glioma

Glioma is one of the most common types of brain tumors [76]. Studies have shown that compared to normal tissues and cell lines, MIR4435-2HG is upregulated in glioma tissues and cell lines. It regulates the expression of genes associated with metabolic pathways, such as aldo-keto reductase family 1 member B1 (AKR1B1), which mediates the cytotoxicity of 2-deoxyglucose, thereby promoting the progression of glioma [40]. Moreover, MIR4435-2HG functions as a sponge to absorb miR-1224-5p, inhibiting its expression and consequently enhancing the expression of transforming growth factor beta receptor 2 (TGFBR2), thus driving the advancement of glioma. Experimental findings have demonstrated that the suppression of MIR4435-2HG can impede the growth and invasion of glioma cells, thereby hindering the progression of tumors in vivo [41]. Another set of experimental outcomes suggests that MIR4435-2HG can modulate the expression of TAZ by functioning as a sponge for miR-125a-5p, thus fostering the advancement of glioma [42]. Additionally, MIR4435-2HG might also enhance the expression of CD44 by absorbing miR-125a-5p and miR-125b-5p, impacting EMT and the TNF-$\alpha$ signaling pathway [43]. These discoveries emphasize the significant role played by MIR4435-2HG in the development of glioma, establishing a basis for its potential efficacy in glioma treatment.

## 2.8. MIR4435-2HG and Head and Neck Squamous Cell Carcinoma

In 2020, head and neck squamous cell carcinoma ranked eighth in the global incidence of malignancies and twelfth in mortality, with approximately 840,000 new cases worldwide. It is projected that by 2030, the number of new cases will increase to around 1 million [61]. Head and neck squamous cell carcinoma is characterized by a high degree of immune deficiency. Due to the subtle early symptoms of the disease and the lack of public awareness about head and neck cancer, it often goes unnoticed, leading to most patients being diagnosed in the advanced stages. In some cases, distant metastasis has already occurred, and two-thirds of the patients face disease progression, with a five-year survival rate of less than 50%. For locally advanced head and neck squamous cell carcinoma patients receiving standard treatment, the rates of local recurrence are approximately 50–60%, while the rates of distant metastatic recurrence range from 4% to 26% [77]. Unfortunately, there are limited treatment options for recurrent or metastatic head and neck squamous cell carcinoma, often resulting in a total survival time of less than a year. Notably, research suggests a significant

presence of MIR4435-2HG in tissues affected by head and neck squamous cell carcinoma, which is strongly linked to the facilitation of cell migration and invasion. Experimental data support the notion that suppressing MIR4435-2HG greatly hinders the ability of cells to migrate and invade [44,45]. Individuals with elevated levels of MIR4435-2HG demonstrate significantly worse prognosis in contrast to those with lower levels, indicating the potential of MIR4435-2HG as a promising biomarker for head and neck squamous cell carcinoma. This biomarker can be utilized for prognosis prediction and precision medicine [46]. These investigations open up novel avenues and potentialities for the treatment and control of head and neck squamous cell carcinoma.

Approximately 80–90% of oral cancer cases and more than half of head and neck squamous cell carcinomas are attributed to oral squamous cell carcinoma, which has a survival rate of 50% after 2 years in advanced stages [78]. Recent studies indicate a notable increase in MIR4435-2HG levels in the blood of patients with oral squamous cell carcinoma. The overexpression of MIR4435-2HG significantly boosts the migratory, proliferative, and invasive capabilities of cells in oral squamous cell carcinoma [47]. Additionally, *Fusobacterium nucleatum*, a bacterium that is notably abundant in oral squamous cell carcinoma tissues compared to normal tissues, has been found to enhance the EMT process by increasing the levels of MIR4435-2HG in oral epithelial cells [48]. Analysis using univariate Cox regression for patients with oral cancer indicates a strong correlation between MIR4435-2HG and patient prognosis, suggesting its potential as a prognostic marker for oral cancer [49]. These discoveries highlight the significant role played by MIR4435-2HG in oral squamous cell carcinoma, offering new perspectives for the treatment and assessment of prognosis in oral cancer.

### 2.9. MIR4435-2HG and Ovarian Cancer

Ovarian cancer, a prevalent malignancy in gynecology, has a rising incidence, poor prognosis, and high mortality rates [79,80]. Recent studies have revealed a notable increase in MIR4435-2HG levels in ovarian cancer, which is closely linked to tumor metastasis [50]. Experimental data demonstrates that suppressing MIR4435-2HG expression effectively hinders the proliferation, invasion, and migration of ovarian cancer cells [51]. Additionally, elevated MIR4435-2HG expression levels can differentiate early-stage (stages I and II) ovarian cancer patients from healthy women, making it a potential diagnostic marker and a promising target for early diagnosis and treatment [50]. This indicates the possibility of utilizing MIR4435-2HG as an innovative focus for treating cervical cancer.

### 2.10. MIR4435-2HG and Cervical Cancer

Among female cancers worldwide, cervical cancer is a prevalent malignancy, ranking fourth [61]. In cervical cancer tissues and cells, there is a notable increase in the expression of MIR4435-2HG. Experiments conducted in a laboratory setting have shown that the decrease in MIR4435-2HG levels can hinder the proliferation, migration, and invasion capabilities of cervical cancer cells [10]. This suggests the potential for MIR4435-2HG to serve as a novel target for cervical cancer treatment.

### 2.11. MIR4435-2HG and Clear Cell Renal Cell Carcinoma

Renal cell carcinoma is one of the most common malignant tumors in the urinary system, accounting for about 4% of adult malignancies [81]. Clear cell renal cell carcinoma, which arises from the epithelial cells of the proximal renal tubules, is a prevalent type of renal cell carcinoma, accounting for approximately 75–80% of renal cell carcinoma cases [82]. The onset of this type of carcinoma is often gradual and it progresses rapidly, leading to frequent diagnosis at an advanced stage [83]. In clear cell renal cell carcinoma specimens, the expression of MIR4435-2HG is markedly increased when compared to adjacent non-cancerous tissue samples. Increased levels of MIR4435-2HG facilitate the growth and movement of cell lines associated with clear cell renal cell carcinoma, whereas decreased MIR4435-2HG hinders these alterations [52].

*2.12. MIR4435-2HG and Bladder Cancer*

Bladder cancer is one of the most common malignant tumors in the urinary system and ranks as the fourth most common cancer in males [84]. In the analysis of TCGA and Gene Expression Omnibus (GEO) dataset, MIR4435-2HG was identified as the only lncRNA with proven prognostic value. Joint validation of MIR4435-2HG was conducted using multiple data sets, and the results demonstrated its significant stratifying effect on bladder cancer prognosis, a finding robustly validated [85]. The expression of MIR4435-2HG in bladder cancer tissue is considerably elevated compared to non-cancerous tissue. Regarding the molecular mechanisms, MIR4435-2HG functions as a sponge that absorbs miR-4288 or miR-2467-3p, thereby enhancing the proliferation, migration, and invasion of cells in bladder cancer [53,54].

*2.13. MIR4435-2HG and Prostate Cancer*

Among men, prostate cancer is the second most prevalent cancerous growth and ranks fifth in terms of fatality [70]. Studies suggest that the plasma levels of MIR4435-2HG in prostate cancer patients are considerably elevated compared to the control group of individuals in good health. The levels of MIR4435-2HG expression in the prostate cancer cell line (PC-3) were found to be elevated, which is associated with a decrease in survival rates of patients with prostate cancer [55]. Furthermore, elevated levels of MIR4435-2HG expression were observed in the prostate cancer cell line (PC-3). Experimental evidence confirms that knockdown of MIR4435-2HG effectively inhibits the proliferation, migration, and invasion capabilities of prostate cancer cells [56].

*2.14. MIR4435-2HG and Melanoma*

Melanoma, a malignant tumor of the skin, is projected to rank as the fifth most prevalent cancer worldwide. It is distinguished by an exceptionally high rate of metastasis and a 5-year survival rate of only 89% [86]. Studies have indicated an increased MIR4435-2HG expression in melanoma stimulates the proliferation and migration of melanoma cells. In terms of mechanism, MIR4435-2HG functions as a "sponge" for miR-802, leading to a decrease in miR-802 levels. Consequently, this results in a rise in the level of expression of flotillin-2 (FLOT2), a protein associated with lipid rafts, consequently enhancing the malignant actions of melanoma cells [57].

## 3. MIR4435-2HG-Related Molecular Mechanisms of Carcinogenesis

MIR4435-2HG is an essential lncRNA that has a substantial impact on the progression of cancer. Playing a crucial role in controlling diverse signaling pathways and cellular biological processes, it has a profound influence on the development and advancement of cancer. Exploring the pathways related to MIR4435-2HG can enhance our comprehension of the molecular mechanisms of cancer, offering fresh perspectives on the diagnosis, treatment, and prevention of this ailment (Figure 1). Thorough examinations of MIR4435-2HG cannot just clarify its exact function in the cancer process but also establish the groundwork for the creation of novel treatment approaches and medications. In the coming years, we can anticipate the emergence of additional therapeutic strategies centered around MIR4435-2HG, offering increased optimism and potential for individuals battling cancer.

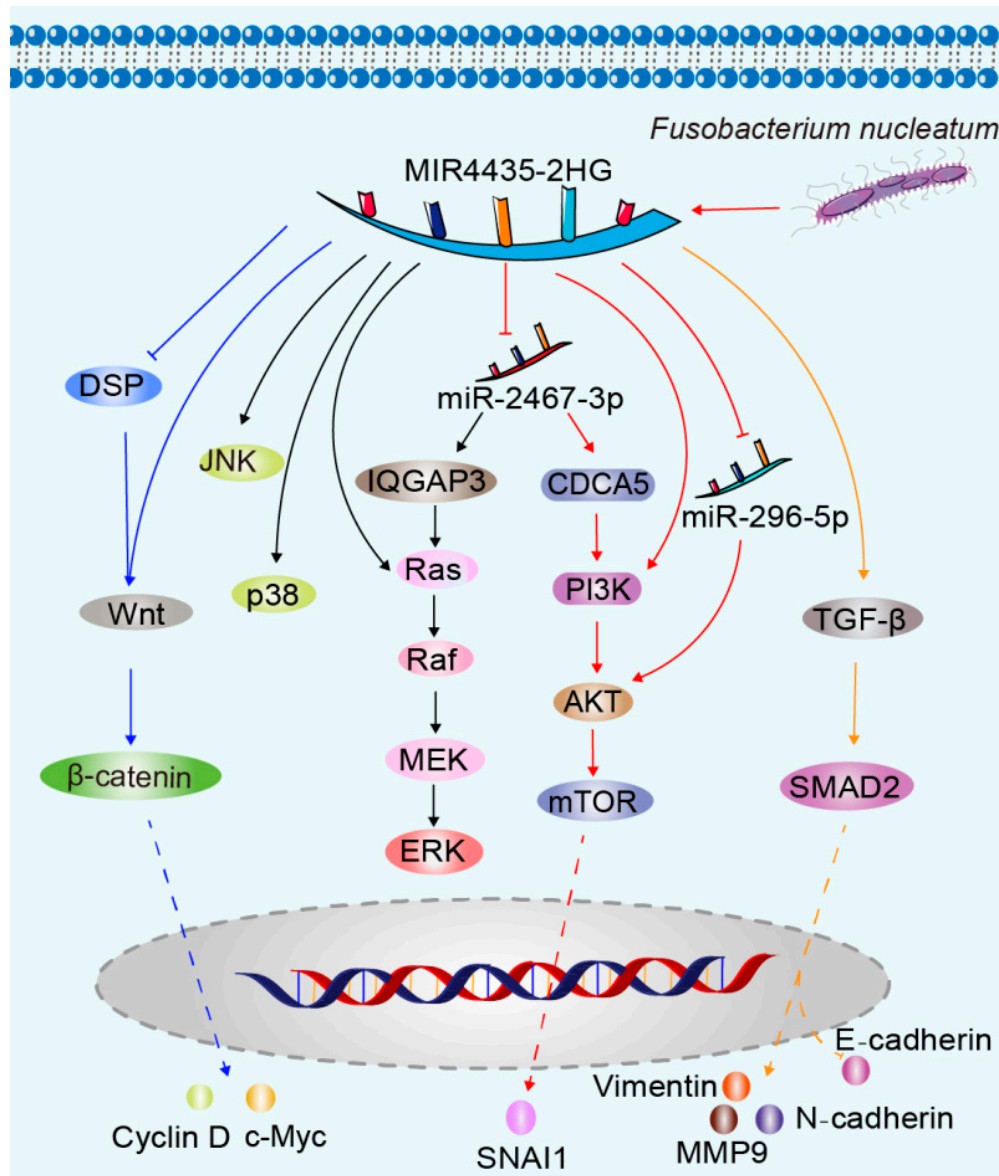

**Figure 1.** LncRNA MIR4435-2HG is involved in signaling pathways for cancer progression. MIR4435-2HG promotes c-Myc, and cyclin D expression through inhibition of DSP or direct activation of Wnt/β-catenin signaling (blue arrow). MIR4435-2HG acts as a sponge to absorb miR-2467-3p, promoting IQGAP3 expression and activating the Ras/Raf/MEK/ERK signaling pathway (black arrow). Additionally, MIR4435-2HG increases the levels of p38 and JNK proteins (black arrow). By sponge-absorbing miR-2467-3p, MIR4435-2HG enhances CDCA5 expression, activates the PI3K/AKT/mTOR signaling pathway, and promotes SNAI1 expression (red arrow). Moreover, *Fusobacterium nucleatum* may enhance the competitive binding of MIR4435-2HG with miR-296-5p, leading to upregulation of AKT protein (red arrow). MIR4435-2HG promotes TGF-β expression, activates SMAD2, increases N-cadherin, MMP9 and vimentin expression, while inhibiting E-cadherin expression (orange arrow).

### 3.1. Wnt/β-Catenin Pathway

The Wnt/β-catenin pathway plays a crucial role in cellular signaling. Abnormal activation of this pathway in cancer can cause an excessive buildup and abnormal movement of β-catenin within the nucleus. This promotes the proliferation, invasion, and metastasis of cancer cells, making them more resistant to cell death. It is closely linked to the development and advancement of various cancers [87–89]. Research has demonstrated that MIR4435-2HG, a lncRNA, plays a vital role, particularly in cancer, by participating in

the control of various signaling pathways and cellular processes. The level of expression is positively associated with the significant increase in β-catenin, particularly in samples of colorectal cancer [90]. MIR4435-2HG may promote the proliferation and inhibit the apoptosis of non-small cell lung cancer cells and ovarian carcinoma cells by activating the Wnt/β-catenin signaling pathway [91,92]. Additionally, it has been confirmed through research that MIR4435-2HG has the ability to activate the Wnt/β-catenin signaling pathway by suppressing the expression of desmoplakin (DSP), thus promoting the growth and metastasis of gastric cancer [12]. Moreover, it has been discovered by scientists that MIR4435-2HG can boost the expression of downstream genes in the Wnt signaling pathway, including β-catenin, c-Myc, and cyclin D [42]. By reducing the presence of MIR4435-2HG, the Wnt/β-catenin signaling pathway can be deactivated, leading to the inhibition of lung cancer and breast cancer cell proliferation and migration [9,36]. The connection between MIR4435-2HG and the Wnt/β-catenin pathway is highly intricate and varied. Exploring further into this connection holds great significance in comprehending the molecular mechanisms of cancer and formulating therapeutic objectives. Further investigation in the future is needed to explore the correlation between MIR4435-2HG and the Wnt/β-catenin pathway, with the goal of offering novel approaches and tactics for the treatment of cancer.

### 3.2. MAPK Pathway

Cancer development heavily relies on the vital involvement of the mitogen-activated protein kinase (MAPK) pathway. The involvement of this signaling pathway in the regulation of various biological processes like cell growth, development, cell death, and invasion is closely linked to the occurrence, progression, metastasis, and prognosis of cancer [93]. Additionally, lncRNAs have a notable impact on the modulation of this signaling pathway [94]. In particular, the activation of the MAPK/ERK pathway by MIR4435-2HG stimulates the growth, progression through the cell cycle, and survival of hepatocellular carcinoma cells. Reducing MIR4435-2HG decreases the phosphorylation levels of ERK, p38, and JNK proteins in hepatocellular carcinoma cells, thereby inhibiting their biological functions [95]. Moreover, the suppression of MIR4435-2HG enhances the levels of miR-2467-3p, resulting in reduced expression of IQ motif GTPase activating protein 3 (IQGAP3). Reducing IQ-GAP3 inhibits the quantities of epidermal growth factor receptor (EGFR) and rat sarcoma virus (Ras), hastening the breakdown of proteins such as rapidly accelerated fibrosarcoma (Raf), mitogen-activated protein kinase (MEK), and extracellular signal-regulated kinase (ERK). It is evident that IQGAP3 triggers the Ras/Raf/MEK/ERK signaling pathway, promoting the advancement of bladder cancer [54]. A thorough comprehension of the association between the MAPK pathway and cancer is crucial for the advancement of more accurate and efficient methods for treating cancer.

### 3.3. PI3K/AKT/mTOR Pathway

The PI3K/AKT/mTOR pathway is an extremely crucial intracellular signaling pathway that has a major regulatory function in the survival, proliferation, growth, and metabolism of cells. The abnormal activation of this pathway is strongly linked to the occurrence, progression, and resistance to treatment of various cancers [96]. MIR4435-2HG plays a crucial role in this pathway by functioning as a sponge for miR-2467-3p, thereby enhancing the expression of cell division cycle-associated 5 (CDCA5), a protein that is involved in the cell division cycle. By promoting the PI3K/AKT/mTOR signaling pathway, this stimulation facilitates the advancement of bladder cancer. Based on experimental findings, it has been demonstrated that the reduction in CDCA5 leads to a notable decrease in the protein levels of p-mTOR, p-PI3K, and p-AKT [54]. Conversely, the silencing of MIR4435-2HG resulted in a significant downregulation of downstream mTOR pathway proteins, such as p-AKT, p-mTOR, p-70S6K, and p-4EBP1. In contrast, the upregulation of MIR4435-2HG amplified the mTOR pathway in these cells, facilitating the EMT in bladder urothelial carcinoma, thereby reinforcing tumor metastasis [15]. Additionally, a separate investigation validated these findings by illustrating that the suppression of MIR4435-2HG

decreased the levels of β-catenin, c-Myc, and cyclin D1. On the other hand, excessive expression of certain molecules such as p-FAK, p-AKT, β-catenin, c-Myc, and cyclin D1 can enhance the ability of prostate cancer cells to proliferate, form colonies, invade, and migrate [56]. This effect is also observed when *Fusobacterium nucleatum* competes with MIR4435-2HG to bind to miR-296-5p, resulting in increased expression of AKT2. Consequently, the expression of the EMT transcription factor snail family transcriptional repressor 1 (SNAI1) is upregulated, promoting the EMT process in oral squamous cell carcinoma cells [48]. In general, the PI3K/AKT/mTOR pathway plays a critical role in the development and progression of cancer, and its abnormal activation is closely associated with cancer cell survival, proliferation, apoptosis, and resistance to drugs. Having a deep comprehension of this pathway holds immense clinical importance in the management and mitigation of cancer.

### 3.4. TGF-β Signaling Pathway

TGF-β, a versatile cytokine, regulates various cellular processes including cell growth, proliferation, differentiation, apoptosis, and extracellular matrix synthesis and degradation [97]. RStudies have demonstrated that inhibiting TGF-β can counteract the migration, proliferation, and invasion induced by overexpressed MIR4435-2HG in oral squamous cell carcinoma cells [47]. Conversely, experimental findings suggest that MIR4435-2HG can enhance the proliferation and migration abilities of non-small cell lung cancer, ovarian cancer, and prostate cancer cells by increasing the expression of TGF-β and activating its signaling pathway [8,50,55]. MIR4435-2HG promotes distant recurrence after resected non-small cell lung cancer by upregulating TGF-β [98]. Additionally, TGF-β inhibitors have been found to reduce the invasion and migration of ovarian cancer and prostate cancer cells [50,55]. Suppression of MIR4435-2HG expression not only inhibits TGF-β expression but also blocks the EMT process in gastric cancer cells by preventing SMAD2 activation. The hindrance is evident in the atypical manifestation of EMT-associated proteins, such as a decline in Twist1 and Snail2 expression, a decrease in N-cadherin and vimentin expression, an increase in E-cadherin expression, and a reduction in matrix metalloproteinase-9 (MMP9) expression [24]. Specifically, MIR4435-2HG propels the advancement of glioma by targeting the miR-1224-5p/TGF-β receptor type 2 (TGFBR2) axis [41]. The stimulation of TGF-β receptors can initiate signal transduction by forming Smad complexes, which have the ability to enter the cell nucleus and function as transcription factors. Furthermore, it has the ability to function via alternative routes apart from Smad pathways, such as the ERK1/2, JNK, and p38 MAPK pathways [99]. MIR4435-2HG might impact cellular proliferation and movement by regulating the TGF-β pathway. Research has identified MIR4435-2HG as a key lnc RNA linked to the TGF-β signaling pathway and the stimulation of hepatic stellate cells. The release of CXCL1 by hepatic stellate cells enhances the expression of TGF-β by absorbing miR-506-3p through MIR4435-2HG sponge, thereby worsening the EMT process in hepatocellular carcinoma cells and increasing their ability to migrate and invade [100]. The impact of TGF-β on cancer is influenced by multiple factors such as the type of tumor, stage of the disease, mutation status of tumor cells, and the tumor microenvironment. Consequently, the TGF-β signaling pathway has emerged as a popular subject of investigation and a promising candidate for cancer therapy. There is potential in modulating the function of TGF-β to inhibit or reverse the formation and progression of tumors.

### 3.5. Epigenetic Regulation

MIR4435-2HG is essential in cancer as it engages in diverse epigenetic regulatory mechanisms that influence gene expression and cellular destiny, ultimately influencing the development, advancement, and management of cancer. Research indicates that there could be a connection between MIR4435-2HG and atypical gene methylation patterns [43,101]. These findings have also been confirmed in ulcerative colitis tissues [102]. However, additional comprehensive investigation is needed to understand the precise regulatory mechanisms involved. In addition to methylation, various other epigenetic regulatory mechanisms, including alterations in histone (e.g., acetylation, methylation, ubiquitination,

phosphorylation) and chromatin restructuring, significantly contribute to the initiation and progression of cancer [103]. The molecular pathways connecting MIR4435-2HG with epigenetic regulation have not been investigated yet. Exploring the function of MIR4435-2HG in the field of epigenetics can lead to a more profound comprehension of the underlying mechanisms that initiate cancer, presenting fresh opportunities and approaches for the diagnosis and treatment of cancer.

### 3.6. Competitive Endogenous RNAs

MIR4435-2HG, as a lncRNA, acts as a "sponge" for miRNAs, binding to miRNA molecules and regulating their stability and function, thus affecting miRNA regulation of their target genes. Current research shows that MIR4435-2HG can sequester miR-138-5p, upregulating SOX4, thereby promoting the growth, migration, and EMT process of gastric cancer cells [26]. In hepatocellular carcinoma, MIR4435-2HG sequesters miR-1-3p, miR-125b-5p, miR-22-3p, miR-513a-5p and miR-136-5p, upregulating downstream target genes such as MM9, SRY-Box transcription factor 12 (SOX12), Ribonucleotide reductase M1/2 subunit (RRM1/2), Tyrosine 3-monooxygenase/tryptophan 5-monooxygenase activation protein zeta (YWHAZ), Kruppel like factor 5 (KLF5), and B3GNT, thereby promoting hepatocellular carcinoma progression [13,27,28,52,104–107]. Additionally, MIR4435-2HG is associated with the upregulation of specific miRNAs, such as the promotion of hepatocellular carcinoma cell proliferation through the upregulation of miRNA-487a [14]. Moreover, MIR4435-2HG, by reducing miR-206 and miR-125b-5p and upregulating Yes-associated protein 1 (YAP1) and Semaphorins 4D (Sema4D), promotes proliferation, migration, invasion, and EMT of colorectal cancer cells [11,35]. It also enhances the vitality, proliferation, migration, invasion, and EMT process of breast cancer cells by increasing transmembrane protein 9B (TMEM9B) expression through the reduction in miR-22-3p levels [39]. By competitively binding to miR-383-5p, MIR4435-2HG increases the protein expression of RNA binding motif 3 (RBM3), affecting the proliferation, migration, and EMT process of head and neck squamous cell carcinoma cells [45]. Furthermore, it triggers the progression of ovarian and cervical cancers by sequestering miR-128-3p and promoting the expression of cyclin-dependent kinase 14 (CDK14) and RNA-binding protein Musashi-2 (MSI2) [10,51]. Another mode of action for MIR4435-2HG is competitive binding to miR-802, which leads to increased expression of FLOT2 and promotes the progression of melanoma [57]. In non-small cell lung cancer, MIR4435-2HG acts as a carcinogen by inhibiting the expression of miR-6754-5p [108]. The molecular mechanism of MIR4435-2HG as a competitive endogenous RNA is summarized in Figure 2. These mechanisms highlight the importance of MIR4435-2HG as a competitive endogenous RNA that influences cancer development by affecting miRNA function and expression, thereby impacting various cancers thorough investigation into the relationship between MIR4435-2HG and miRNAs is essential to gain a deeper comprehension of the initiation and advancement of cancer.

### 3.7. Effects on Immune Cells

MIR4435-2HG may play a crucial role in the tumor immune microenvironment, particularly in close association with immune cell infiltration in tumors. The infiltration status and activity of immune cells play a pivotal role in tumor development and treatment. MIR4435-2HG might regulate immune cell function and infiltration in various ways, thereby influencing tumor growth and metastasis [109,110]. MIR4435-2HG is present in exosomes, and these exosomes activate the Jagged1 (JAG1)/Notch pathway and Janus kinase 1 (JAK1)/signal transducer and activator of transcription 3 (STAT3) pathway, promoting the polarization of M2 macrophages, enhancing the EMT process, and migration of gastric cancer cells [25,111]. The β-chain cytokines (IL-3, IL-5, and GM-CSF) inhibit the expression of Bcl2l11, also referred to as Bim, a gene that promotes cell apoptosis, in a manner dependent on MIR4435-2HG. This regulation is of utmost importance for the survival of mature eosinophils, neutrophils, and monocytes [112]. Radiation causes a sharp decrease in peripheral blood lymphocyte count and a temporary increase in granulocyte

count, leading to a relative increase in neutrophils and other cells in the whole blood. MIR4435-2HG has the potential to be developed as a biomarker for assessing radiation biodosimetry [113]. In another experiment, IL-6 induced the overactivation of the Src homology region 2-containing protein tyrosine phosphatase 2 (SHP2)-STAT3 signaling axis, leading to increased expression of the novel anti-apoptotic lncRNA MIR4435-2HG in Tet2-KO myeloid cells and hematopoietic stem and progenitor cells. Genetic loss of MIR4435-2HG could rescue inflammatory stress-induced abnormalities in hematopoietic stem and progenitor cells and mature bone marrow cells [114]. However, Kotzin et al. later discovered that the activation of MIR4435-2HG can promote the expression of the pro-apoptotic factor BCL2L11 in CD8+ T cells, which is detrimental to the survival of CD8+ T cells following lymphocytic choriomeningitis virus infection. MIR4435-2HG suppresses the expression of IFN-γ, TNF-α, and granzyme B by inhibiting the PI3K-AKT signaling pathway [115]. The role of MIR4435-2HG in the tumor immune microenvironment is diverse and complex. It might influence the tumor immune status by regulating immune-related genes, affecting immune cell infiltration and activity, among other ways. In-depth research into its mechanisms is expected to provide new targets and strategies for tumor immunotherapy.

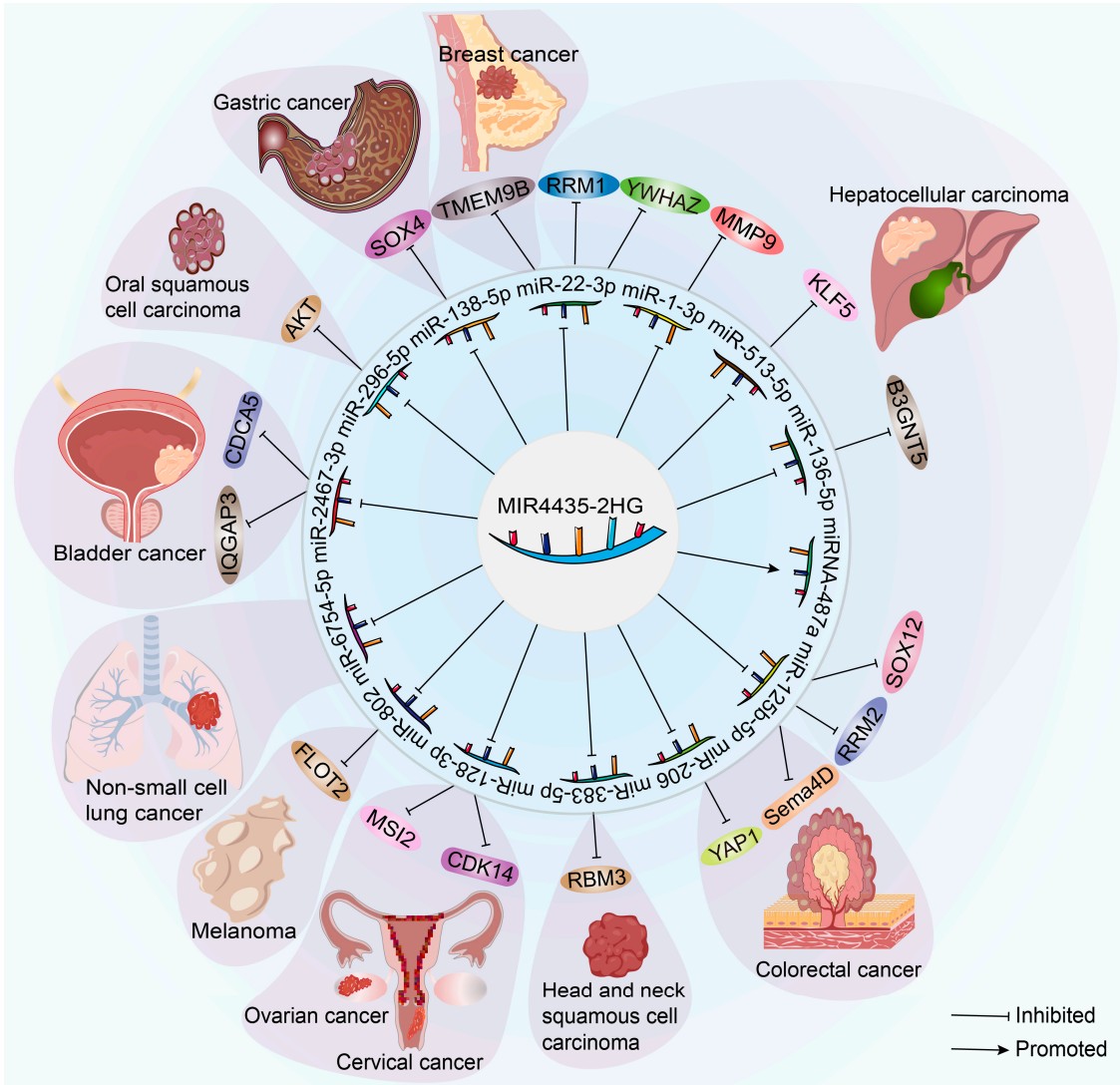

**Figure 2.** LncRNA MIR4435-2HG acts as a ceRNA in several human cancers and exerts oncogenic effects by sponging miRNAs.

Furthermore, MIR4435-2HG may be involved in certain cell death processes, such as ferroptosis, pyroptosis, and necroptosis [46,116,117]. MIR4435-2HG potentially participates in cell death processes by regulating specific signaling pathways, gene expression, or cellular signaling molecules. It may impact apoptosis or other forms of cell death and could also be related to pathways associated with cell survival. However, further in-depth research is needed to reveal the specific role and mechanisms of MIR4435-2HG in these processes.

## 4. Conclusions and Perspectives

In summary, we can observe that its abnormal expression as a lncRNA is closely associated with the occurrence and progression of malignant tumors in various cancers. Research shows that MIR4435-2HG is involved in regulating multiple biological processes in cancer cells, such as proliferation, invasion, metastasis, and drug resistance. This impact on tumor development is achieved through interactions with cell cycle regulation, epigenetic modifications, signaling pathways, and more.

MIR4435-2HG can be considered a potential therapeutic target or diagnostic biomarker. Intervention strategies targeting MIR4435-2HG, such as small molecule drugs, antibodies, nucleic acid drugs, and others, could potentially pave the way for new approaches in cancer treatment. By intervening with MIR4435-2HG and its related signaling pathways, there is potential for precise cancer treatment, improving therapeutic effectiveness, and reducing toxic side effects.

However, current research is still in its early stages and has certain limitations. One of the primary limitations of the study is the reliance on available clinical data, which may not capture the full spectrum of MIR4435-2HG's role in different tumor types. Clinical data might be subject to biases, and future studies should aim to collect more comprehensive and diverse datasets. The studies may be limited by the sample size and the lack of diversity in the patient population. Expanding the study with larger and more diverse cohorts can provide a more robust understanding of MIR4435-2HG's role in tumors. MIR4435-2HG's effects can be tissue-specific, and existing research may not cover all possible tumor types. Future investigations should explore its role in a wider range of cancers and different tissues. The evaluation methodologies for MIR4435-2HG expression and function can vary among different studies. Standardizing methodologies and using common platforms for assessment will enhance the comparability of research outcomes. While accumulating studies have uncovered several mechanisms related to MIR4435-2HG, more in-depth studies are needed to dissect its precise functional roles in tumor development and progression. This might involve exploring specific signaling pathways and interaction partners. Importantly, treatment strategies targeting MIR4435-2HG are still under research, and clinical applications will take time to materialize.

In the future, we look forward to more in-depth research revealing the biological functions and mechanisms of MIR4435-2HG, accelerating its clinical applications in the field of cancer. Moreover, interdisciplinary collaboration will enhance our understanding of the role of MIR4435-2HG and its potential applications in cancer treatment, providing more choices and better treatment outcomes for cancer patients.

**Author Contributions:** Z.C., D.G. and W.Z. conceived the review. Z.C., D.G., Q.Z., Z.W., F.H. and W.Z. undertook the initial research. Z.C. and D.G. were involved in writing, W.Z. reviewed the manuscript. D.G. contributed equally to this work and should be considered co-first author. All authors have read and agreed to the published version of the manuscript.

**Funding:** This article was supported by Natural Science Foundation of Gansu Province (22JR11RA033), Lanzhou Science and Technology Plan Project (2023-2-75), The First Hospital of Lanzhou University Excellent Doctoral Research Initiation Fund (ldyyyn2021-114), The First Hospital of Lanzhou University Intra-Hospital Fund Youth Fund (ldyyyn2022-38).

**Institutional Review Board Statement:** Not applicable.

**Data Availability Statement:** Not applicable.

**Conflicts of Interest:** The authors declare no conflict of interest.

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
