# Peer review of "Biological Roles and Pathogenic Mechanisms of LncRNA MIR4435-2HG in Cancer: A Comprehensive Review"

_cimb, doi:10.3390/cimb45110556_

Round 1

Reviewer 1 Report

Comments and Suggestions for Authors

The authors have meticulously gathered evidence showcasing abnormal regulation of MIR4435-2HG in various cancer types including stomach, liver, colon, lung, breast, and ovarian cancers. They further delineate how overexpression correlates with tumor progression characteristics such as tumor size, staging, metastasis, and poor prognosis. Experimental inhibition of MIR4435-2HG suggests its role in promoting cancer cell growth, invasion, and metastasis.

The discussion on molecular mechanisms is indeed enlightening. Authors elucidate how MIR4435-2HG interacts with key signaling pathways like Wnt/β-catenin, MAPK, PI3K/AKT/mTOR, and TGF-beta, thus promoting cancer. Other notable mechanisms include epigenetic regulation and competing endogenous RNA activity. Through an exhaustive literature review, a spectrum of cancer-associated functions of MIR4435-2HG has been highlighted.

Overall, this is an excellent review encapsulating current knowledge on a significant long non-coding RNA in cancer biology. The authors have done a commendable job summarizing a vast amount of literature. I believe this review makes a strong contribution and will serve as a useful reference for researchers in this field. The potential of targeting MIR4435-2HG as a therapeutic strategy is well acknowledged, paving the way for future directions. I highly recommend the publication of this review. To further contribute to the understanding of non-specialist readers and to the advancement of the research field, I suggest the following:

Comment

#1. I recommend adding a section in the introduction regarding the potential therapeutic targeting of MIR4435-2HG. For instance, discussing strategies like siRNA, antisense oligonucleotides, and small molecules, and how they could be utilized based on the explained mechanisms would aid reader comprehension.

#2. Research Limitations:In the summary section, please provide a discussion on the potential limitations of the research concerning MIR4435-2HG. Mentioning the possibilities of in vivo functional studies and evaluation methodologies on common platforms could further assist reader understanding.

Author Response

Dear Editors and Reviewers:
We deeply appreciate that you have spent your valuable time reviewing our manuscript, and we are very grateful for the reviewers’ comments concerning our manuscript entitled Biological Roles and Pathogenic Mechanisms of LncRNA MIR4435-2HG in Cancer: A Comprehensive Review”. Those comments are all valuable and very helpful for revising and improving our paper, as well as the important guiding significance to our researches. We have studied comments carefully and have made correction which we hope meet with approval (marked in red). If there is any missing, wrong and unsatisfactory content, please feel free to contact us, we would appreciate your valuable comments and would be happy to further improve our review.

Reviewer #1:

#1. I recommend adding a section in the introduction regarding the potential therapeutic targeting of MIR4435-2HG. For instance, discussing strategies like siRNA, antisense oligonucleotides, and small molecules, and how they could be utilized based on the explained mechanisms would aid reader comprehension.

Thank you for your valuable suggestion. We have added the following section to the introduction. Utilizing the oncogenic potential of MIR4435-2HG in various cancers, we employ short interfering RNA (siRNA) and antisense oligonucleotides (ASO) to facilitate oncogene suppression and reduce the expression of target gene proteins, thereby yielding therapeutic benefits. Considering the intricate spatial structures and multifaceted biological functions of lncRNAs in cancer therapy, lncRNAs have arisen as promising drug targets for modulating several small molecules. Nevertheless, the development of small molecule drugs targeting lncRNAs is still in the pre-clinical research phase, and there have been no reported instances of small molecule drugs targeting lncRNAs advancing to clinical trials. Regardless, investigating the correlation between MIR4435-2HG and cancer can assist in gaining a more profound comprehension of disease mechanisms and offering hints for innovative therapeutic methods.

#2. Research Limitations:In the summary section, please provide a discussion on the potential limitations of the research concerning MIR4435-2HG. Mentioning the possibilities of in vivo functional studies and evaluation methodologies on common platforms could further assist reader

Thank you for your valuable suggestion. We have added the following section to the introduction. However, current research is still in its early stages and has certain limitations. One of the primary limitations of the study is the reliance on available clinical data, which may not capture the full spectrum of MIR4435-2HG's role in different tumor types. Clinical data might be subject to biases, and future studies should aim to collect more comprehensive and diverse datasets. The studies may be limited by the sample size and the lack of diversity in the patient population. Expanding the study with larger and more diverse cohorts can provide a more robust understanding of MIR4435-2HG's role in tumors.  MIR4435-2HG's effects can be tissue-specific, and existing research may not cover all possible tumor types. Future investigations should explore its role in a wider range of cancers and different tissues. The evaluation methodologies for MIR4435-2HG expression and function can vary among different studies. Standardizing methodologies and using common platforms for assessment will enhance the comparability of research outcomes. While accumulating studies have uncovered several mechanisms related to MIR4435-2HG, more in-depth studies are needed to dissect its precise functional roles in tumor development and progression. This might involve exploring specific signaling pathways and interaction partners. Importantly, treatment strategies targeting MIR4435-2HG are still under research, and clinical applications will take time to materialize.

Once again, we are very grateful for the reviewers’ comments concerning our manuscript entitled Biological Roles and Pathogenic Mechanisms of LncRNA MIR4435-2HG in Cancer: A Comprehensive Review”. If there is any missing, wrong and unsatisfactory content, please feel free to contact us, we would appreciate your valuable comments and would be happy to further improve our review.

Sincerely yours,

Wence Zhou

Reviewer 2 Report

Comments and Suggestions for Authors

Chen et al. wrote an extensive review discussing the biological roles and pathogenic mechanisms of the long non-coding RNA (LncRNA) MIR4435-2HG in cancer. MIR4435-2HG is involved in various biological processes, including energy metabolism, immune responses, cell cycle regulation, and apoptosis. Recent research has revealed its positive association with different types of cancers, suggesting its potential as a target for cancer therapy.

However, the review's structure and content may not be novel enough when compared to previous publications (e.g. Zhang et al., Biomed Pharmacother 2022; Zhao et al., Curr Pharm Des . 2022). It is recommended that the authors focus more on the molecular mechanisms, especially in areas that have not been covered by earlier reviews (e.g. epigenetic regulations and effects on immune cells). Additionally, the authors briefly mention that MIR4435-2HG can act as a tumor suppressor in some studies, and it's suggested that they expand on this aspect.

Author Response

Dear Editors and Reviewers:
We deeply appreciate that you have spent your valuable time reviewing our manuscript, and we are very grateful for the reviewers’ comments concerning our manuscript entitled Biological Roles and Pathogenic Mechanisms of LncRNA MIR4435-2HG in Cancer: A Comprehensive Review”. Those comments are all valuable and very helpful for revising and improving our paper, as well as the important guiding significance to our researches. We have studied comments carefully and have made correction which we hope meet with approval (marked in red). If there is any missing, wrong and unsatisfactory content, please feel free to contact us, we would appreciate your valuable comments and would be happy to further improve our review.

Thank you for your thoughtful feedback and for highlighting the previous reviews by Zhang et al., and Zhao et al. We appreciate your insights and agree that it's essential to provide a fresh perspective and novel insights in our review. 

In response to your suggestions, we have focused on elaborating the molecular mechanisms of MIR4435-2HG, particularly in areas that haven't been comprehensively covered by earlier reviews, such as epigenetic regulations and its effects on immune cells. Our aim is to expand the research content on MIR4435-2HG in these aspects as much as possible to ensure the uniqueness of our review.

Regarding the mention of MIR4435-2HG acting as a tumor suppressor in some studies, there is currently limited literature available on this aspect. However, Kotzin et al. later discovered that the activation of MIR4435-2HG can promote the expression of the pro-apoptotic factor BCL2L11 in CD8+ T cells, which is detrimental to the survival of CD8+ T cells following lymphocytic choriomeningitis virus infection. MIR4435-2HG suppresses the expression of IFN-γ, TNF-α, and granzyme B by inhibiting the PI3K-AKT signaling pathway [115]. With further in-depth research on MIR4435-2HG, additional related mechanisms may be discovered.

Once again, we are very grateful for the reviewers’ comments concerning our manuscript entitled Biological Roles and Pathogenic Mechanisms of LncRNA MIR4435-2HG in Cancer: A Comprehensive Review”. If there is any missing, wrong and unsatisfactory content, please feel free to contact us, we would appreciate your valuable comments and would be happy to further improve our review.

Sincerely yours,

Wence Zhou

Reviewer 3 Report

Comments and Suggestions for Authors

This review article concentrates on the role of lncRNA MIR4435-2HG in cancer. The introduction describes the general functions of this lncRNA. Subsequently, the authors present data concerning MIR4435-2HG in many kinds of cancer. They showed a general up-regulation of this nucleic acid. The second part of the manuscript concentrates on molecular mechanisms connecting MIR4435-2HG and cancer development, e.g., the role of Wnt/beta-catenin, MAPK, or TGF-beta pathways, as well as epigenetics and immune regulation.

In my opinion, the article by Chen et al. is interesting and may be helpful for readers. However, I suggest some corrections:

1. Paragraph 2.8 describes head and neck squamous cell carcinoma data, and 2.9.  oral SCC data. The second one is a part of HNSCC. Thus, these paragraphs should be unified as one paragraph.

2. The same suggestion concerning Table 1 and HNSCC + OSCC.

3. In line 199, the authors wrote information about HNSCC incidence per year. This data comes from the USA population, not a global summary. Please indicate it or replace the data with international statistics.

4. Figure 2 must also be cited in the main text.

Author Response

Dear Editors and Reviewers:
We deeply appreciate that you have spent your valuable time reviewing our manuscript, and we are very grateful for the reviewers’ comments concerning our manuscript entitled Biological Roles and Pathogenic Mechanisms of LncRNA MIR4435-2HG in Cancer: A Comprehensive Review”. Those comments are all valuable and very helpful for revising and improving our paper, as well as the important guiding significance to our researches. We have studied comments carefully and have made correction which we hope meet with approval (marked in red). If there is any missing, wrong and unsatisfactory content, please feel free to contact us, we would appreciate your valuable comments and would be happy to further improve our review.

We appreciate your valuable comments and apologize for the distress caused by our imprecise expression and misspelling. We carefully revised based on each comment (marked in red).
1. Paragraph 2.8 describes head and neck squamous cell carcinoma data, and 2.9. oral SCC data. The second one is a part of HNSCC. Thus, these paragraphs should be unified as one paragraph.
We appreciate your corrections to our inaccuracies. We have consolidated some of the content into "2.8. MIR4435-2HG and Head and Neck Squamous Cell Carcinoma."
2. The same suggestion concerning Table 1 and HNSCC + OSCC.
In Table 1, we have also summarized the corresponding sections as "Head and neck squamous cell carcinoma".
3. In line 199, the authors wrote information about HNSCC incidence per year. This data comes from the USA population, not a global summary. Please indicate it or replace the data with international statistics.
Thank you for your correction. We have addressed your concern by replacing the previous information in line 199 with the following revised statement: “In 2020, head and neck squamous cell carcinoma ranked eighth in the global incidence of malignancies and twelfth in mortality, with approximately 840,000 new cases worldwide. It is projected that by 2030, the number of new cases will increase to around 1 million.” We have made the necessary changes to accurately reflect the global incidence of head and neck squamous cell carcinoma.
4.Figure 2 must also be cited in the main text.
Thank you for your reminder. We have incorporated the necessary citation of Figure 2 in the main text in accordance with your suggestion. Please refer to "The molecular mechanism of MIR4435-2HG as a competitive endogenous RNA is summarized in Figure 2."
Once again, we are very grateful for the reviewers’ comments concerning our manuscript entitled Biological Roles and Pathogenic Mechanisms of LncRNA MIR4435-2HG in Cancer: A Comprehensive Review”. If there is any missing, wrong and unsatisfactory content, please feel free to contact us, we would appreciate your valuable comments and would be happy to further improve our review.
Sincerely yours,

Round 2

Reviewer 2 Report

Comments and Suggestions for Authors

The authors have made an effort to expand some sections, particularly focusing on the effects of MIR4435-2HG on immune cells. However, the majority of the content and the overall structure of the manuscript still bear significant similarities to existing review articles. Regrettably, it remains challenging to overcome this drawback. Thus, I maintain my reservation about recommending this manuscript for publication. To enhance its suitability for publication, the authors may need to make substantial changes by providing more unique insights or addressing gaps in the existing literature.